# Prevalence of Insomnia in Two Saskatchewan First Nation Communities

**James A Dosman [1,2,\*], Chandima P Karunanayake [1,\*], Mark Fenton [2], Vivian R Ramsden [3], Robert Skomro [2], Shelley Kirychuk [1,2], Donna C Rennie [4], Jeremy Seeseequasis [5], Clifford Bird [6], Kathleen McMullin [1], Brooke P Russell [1], Niels Koehncke [1,2], Thomas Smith-Windsor [7], Malcolm King [8], Sylvia Abonyi [8] and Punam Pahwa [1,8]**

1 Canadian Centre for Health and Safety in Agriculture, University of Saskatchewan, 104 Clinic Place, Saskatoon, SK S7N 2Z4, Canada; shelley.kirychuk@usask.ca (S.K.); kathleen.mcmullin@usask.ca (K.M.); bpr053@mail.usask.ca (B.PR.); niels.koehncke@usask.ca (N.K.); pup165@mail.usask.ca (P.P.)
2 Department of Medicine, University of Saskatchewan, Royal University Hospital, 103 Hospital Drive, Saskatoon, SK S7N 0W8, Canada; mef132@mail.usask.ca (M.F.); r.skomro@usask.ca (R.S.)
3 West Winds Primary Health Centre, Department of Academic Family Medicine, University of Saskatchewan, 3311 Fairlight Drive, Saskatoon, SK S7M 3Y5, Canada; viv.ramsden@usask.ca
4 College of Nursing, University of Saskatchewan, 104 Clinic Place, Saskatoon, SK S7N 2Z4, Canada; donna.rennie@usask.ca
5 Community A, PO Box 96, Duck Lake, SK S0K 1J0, Canada; jccquasis@willowcreehealth.com
6 Community B, PO Box 250, Montreal Lake, SK S0J 1Y0, Canada; c.bird@sasktel.net
7 Victoria Hospital, Prince Albert, SK S6V 4N9, Canada; dr.tom@sasktel.net
8 Department of Community Health & Epidemiology, College of Medicine, University of Saskatchewan, 107 Wiggins Road, Saskatoon, SK S7N 5E5, Canada; malcolm.king@usask.ca (M.K.); sya277@mail.usask.ca (S.A.)
\* Correspondence: james.dosman@usask.ca (J.AD.); cpk646@mail.usask.ca (C.PK.); Tel.: +1-306-966-1475 (J.AD.); +1-306-966-1647 (C.PK.)

**Abstract:** Insomnia is a common problem in Canada and has been associated with increased use of health care services and economic burden. This paper examines the prevalence and risk factors for insomnia in two Cree First Nation communities in Saskatchewan, Canada. Five hundred and eighty-eight adults participated in a baseline survey conducted as part of the First Nations Sleep Health Collaborative Project. The prevalence of insomnia was 19.2% among participants with an Insomnia Severity Index score of ≥15. Following the definition of nighttime insomnia symptoms, however, the prevalence of insomnia was much higher, at 32.6%. Multivariate logistic regression modeling revealed that age, physical health, depression diagnosis, chronic pain, prescription medication use for any health condition, and waking up during the night due to terrifying dreams, nightmares, or flashbacks related to traumatic events were risk factors for insomnia among participants from two Saskatchewan Cree First Nation communities.

**Keywords:** insomnia; nighttime insomnia; First Nations; adults

## 1. Introduction

Insomnia is a common problem in Canada and is associated with increased use of health care services and economic burden [1,2]. A study conducted in Quebec, Canada, reported that the total annual cost of insomnia was $6.6 billion CAD, inclusive of the annual direct and indirect costs [1]. Direct costs were associated with health-care consultations, transportations for these consultations, prescription medication, other over-counter products, and alcohol used as a sleep aid. Indirect costs were associated with insomnia-related absenteeism from work and insomnia-related productivity losses [1]. In addition, insomnia has been shown to significantly contribute to accidents, absenteeism from work, and decreased work productivity [3]. Epidemiological studies around the world including Canada reported that prevalence of insomnia varied from 4% to 50% [4–13]. Many definitions exist

for insomnia, and estimates vary by definition [2,14,15]. Most commonly used definitions for insomnia are based on nighttime insomnia symptoms [8], the Insomnia Severity Index (ISI) score ≥15 [16–19], or a Pittsburgh Sleep Quality Index (PSQI) score >5 [20]. In Canada, reported insomnia prevalence varied from 13.4% to 38% [5,8,11,21,22] depending on the measure used, such as single question (trouble falling or staying asleep), multiple questions (trouble falling or staying asleep and early awakening) or use of a scale (ISI score).

Insomnia is associated with many demographic and proximal social factors such as sex [2,7,11,13,16,21–24], age [2,7,11,13,14,21,22], marital status [2,7,11,13], education [7,11,16], family income [2,7,11,14,16,21]), smoking [13], and consumption of caffeinated beverages [13]. Health outcomes such as mental and psychological health status, anxiety [2,11,13,14,16,22], depression [2,16,22,25,26], as well as chronic health conditions [2,10,14,26–28] including sleep apnea [29,30], kidney disease [31,32], thyroid disorders [33], poor health [2,13,21], and quality of life [21,34] are also associated with insomnia.

Lombardero et al. [35] reported that the insomnia prevalence of North American Indian/Alaska Native populations ranged from 25% to 33%. Taylor et al. [36] reported that insomnia rates were higher among North American Indian/Alaska Native military personnel compared to other ethnic military personnel (33.7% vs. 15.1% to 21.4%). To our knowledge, there is no literature available on the prevalence of insomnia among Indigenous peoples in Canada. There are health inequities between Indigenous and non-Indigenous (mainly descending from Europeans) peoples in Canada [37–39] located in historical, political, social, and economic conditions that inequitably influence Indigenous health. Colonization is a key upstream determinant of health for Indigenous peoples that impacts more proximal determinants of Indigenous health through systemic and individual racism [40–43]. One consequence is that Indigenous youth and adults experience bullying and abuse [44,45] that affect sleep and are linked to the prevalence of insomnia. This paper reports the prevalence and risk factors of insomnia among participants from two Saskatchewan Cree First Nation communities.

## 2. Materials and Methods

### 2.1. Study Sample

The data for this study came from a baseline survey undertaken during the First Nations Sleep Health Project (FNSHP) conducted in partnership with two Cree First Nation communities (Community A and Community B) in Saskatchewan in 2018–2019. First Nations comprise one of the three groups of Indigenous peoples who are the descendants of the original inhabitants of North America (the other two being Inuit and Métis). First Nations peoples have unique heritages, languages, cultural practices, and spiritual beliefs [46,47]. All Indigenous peoples in Canada are impacted by historical circumstances and contemporary contingencies of colonization that produce inequities in social and structural determinants of health [48]. The purpose of the FNSHP is to examine the relationships between sleep disorders and risk factors and co-morbidities, and to evaluate local diagnosis and treatment. The study was approved by the University of Saskatchewan's Biomedical Research Ethics Board (Certificate No. Bio #18-110) and followed Chapter 9 (Research Involving the First Nations, Inuit, and Métis Peoples of Canada) of the Tri-Council Policy Statement: Ethical Conduct for Research Involving Humans [49]. In addition to community consent achieved through the collaboration process, informed written consent was obtained from each individual participant following a discussion of study objectives, procedures, risks, and benefits.

### 2.2. Data Collection

Trained research assistants from each community conducted the baseline surveys in their respective community. Adults 18 years and older were invited to the Community Health and/or Youth Centre to complete the interviewer-administered questionnaires and clinical assessments. A pamphlet describing the study and an invitation to participate were distributed by the research assistants during local community events such as "Treaty

Days" and during door-to-door canvassing. Simultaneously, there was a social media campaign to invite the community members to participate in the survey. The survey collected information on demographic variables, individual and contextual determinants of sleep health, self-reported height and weight, and objective clinical measurements. This manuscript is based on data from the questionnaires. Demographic information about participants including age, sex, body mass index, education level and money left at the end of the month, life-style factors, house environment, medical history, and sleep health information was obtained from the survey questionnaire.

The number of chronic health conditions was derived by adding up positive responses to the following conditions: high blood pressure, heart problems, stroke, high cholesterol and/or triglycerides, diabetes, atrial fibrillation, chronic obstructive pulmonary disease/emphysema, asthma, chronic bronchitis, acid reflux, hypothyroidism, severe eyesight problem, sinus problems, Parkinson's disease, sleep apnea, kidney disease, and restless legs syndrome. Number of chronic health conditions was grouped into four categories: none, one condition, two conditions, and three or more conditions. Other clinical factors considered separately were self-rated physical and mental health status, anxiety, depression, post-traumatic stress disorder, and chronic pain. Finally, prescription medication use for any health condition, sleep medication, traditional medicines to aid in sleeping, and other aids for sleep were considered.

### 2.3. Definitions

2.3.1. Nighttime Insomnia Symptoms

Nighttime insomnia symptoms were determined by the question, "How many days per week do you have trouble going to sleep or staying asleep?" Respondents who answered either "most of the time" or "all of the time" were considered to have nighttime insomnia symptoms, as previously reported [8].

2.3.2. Sleep Duration

Sleep duration was calculated from answers to questions about participant's usual sleep habits during the past month: "When have you usually gone to bed?"; "When have you usually gotten up in the morning?"; "How long has it taken to fall asleep each night?". Taking the difference of first two questions, the time in bed was calculated. The time to falling asleep was subtracted from time in bed to get the actual sleep duration. Recommendations on sleep duration hours were based on those from the National Sleep Foundation [50]. Respondents were classified as sleeping less than the recommended number of hours for optimal health (7 h per night for ages 18 to 79) or sleeping the recommended hours or more [50].

2.3.3. Pittsburgh Sleep Quality Index Score

The Pittsburgh Sleep Quality Index (PSQI) calculating the seven components (subjective sleep quality, sleep latency, sleep duration, habitual sleep efficiency, sleep disturbances, use of sleep medication, and daytime dysfunction over the last month) was determined [51,52]. By adding up the scores from the seven components, a Global PSQI Score was calculated. A Global sum of 5 or greater indicated a "poor" sleep; thus, the participant was then considered to have insomnia [20].

2.3.4. Insomnia Severity Index (ISI)

The ISI has seven self-reported questions assessing the nature, severity, and impact of insomnia [17,19]. The participant was asked to rate the "current" (that is, the last two weeks) severity of their insomnia problems, and the dimensions evaluated were severity of sleep onset, sleep maintenance, early morning awakening problems, sleep dissatisfaction, interference of sleep difficulties with daytime functioning, noticeability of sleep problems by others, and distress caused by the sleep difficulties [19]. A 5-point Likert scale is used to rate each question, yielding a total score ranging from 0 to 28. The total score was

interpreted as follows: absence of insomnia (0–7), sub-threshold insomnia (8–14), moderate insomnia (15–21), and severe insomnia (22–28) [19]. Clinical insomnia was identified if the score was equal to or greater than 15 [18]. The focus of this paper was on clinical insomnia, which is defined by an ISI score ≥15.

### 2.4. Statistical Analysis

Statistical analyses were conducted using SPSS version 27 (IBM Corp. Released 2020. IBM SPSS Statistics for Windows, Version 27.0. Armonk, NY: IBM Corp.). Descriptive statistics, mean, and standard deviation (SD) were reported for continuous variables, and *p*-value of the t-test was reported when comparing the means of two samples. For categorical variables, frequency and percentage (%) were reported. Chi-square tests were used to determine the bivariable association of insomnia prevalence with independent variables of interest. Logistic regression models were used to predict the relationship between a binary outcome of insomnia (yes or no) and a set of explanatory variables. A series of logistic regression models were fitted to determine whether potential risk factors, confounders, and interactive effects contributed significantly to the prediction of insomnia prevalence. Based on bivariable analysis, variables with $p < 0.20$ and less than 25% missing information were candidates for the multivariate model. All variables that were statistically significant ($p < 0.05$), as well as important clinical factors (sex, number of chronic conditions), were retained in the final multivariable model. Interactions between potential effect modifiers were examined and were retained in the final model if the *p*-value was <0.05. The strengths of associations were presented by odds ratios (OR) and their 95% confidence intervals (CI) [53].

### 3. Results

Five hundred and eighty-eight individuals participated in the baseline survey, 418 individuals from Community A and 170 individuals from Community B. The mean age (±SD) of the 588 study participants was 40.0 ± 15.3 years with a range of 18–78 years. Median age was 38 years. There were 44.2% male and 55.8% females that participated in this study. Duration of sleep hours was available for 567 adults.

Table 1 shows the percentage of individuals reporting each item of Insomnia Severity Index. Most frequent higher scored items were Item 1, Item 3, Item 4, Item 5, and Item 6. Nighttime insomnia was defined as trouble going to sleep or staying asleep most of the time (16.2%) or all the time (16.4%) (Figure 1).

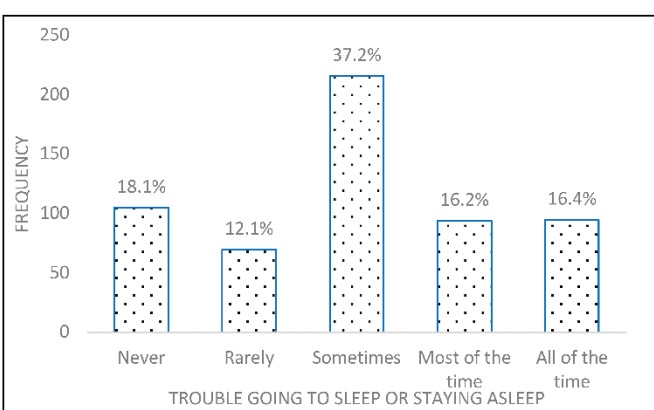

**Figure 1.** Trouble going to sleep or staying asleep.

**Table 1.** Percentage of the sample responded to each item of Insomnia Severity Index (ISI).

| Item ISI | Item Response Choice * | | | | |
|---|---|---|---|---|---|
| | **0** | **1** | **2** | **3** | **4** |
| 1. Falling asleep (*n* = 582) | 30.8 | 25.1 | 29.7 | 11.0 | 3.4 |
| 2. Staying asleep (*n* = 579) | 35.4 | 26.9 | 26.3 | 8.8 | 2.6 |
| 3. Early awakening (*n* = 577) | 34.5 | 21.8 | 25.5 | 13.0 | 5.2 |
| 4. Satisfaction (*n* = 580) | 9.7 | 31.6 | 31.7 | 21.0 | 6.0 |
| 5. Interference (*n* = 578) | 31.0 | 29.6 | 24.7 | 9.2 | 5.5 |
| 6. Noticeable (*n* = 577) | 36.9 | 23.1 | 25.6 | 8.8 | 5.5 |
| 7. Worry (*n* = 581) | 40.4 | 24.3 | 24.1 | 7.6 | 3.6 |

* Items 1–3—0, no problem; 1, mild; 2, moderate; 3, severe; 4, very severe; Item 4—0, very satisfied; 1, satisfied; 2, moderately satisfied; 3, dissatisfied; 4, very dissatisfied; Items 5–7—0, not at all; 1, a little; 2, somewhat; 3, much; 4, very much.

The prevalence of nighttime insomnia symptoms was 32.6% (189/580). According PSIQ score, 65.0% (359/552) reported poor sleep as an indication of insomnia. The prevalence of clinical insomnia was 19.2% (109/567) (Table 2). Average (± SD) sleep hours among insomnia participants were significantly lower when compared with those that did not have insomnia (7.24 ± 2.75 h vs. 8.35 ± 2.07 h; $p < 0.0001$). Participants who had insomnia were significantly more likely to have less than the recommended number of hours of sleep (<7 h) compared to those that did not have insomnia (44.8% vs. 21.4%; $p < 0.0001$).

**Table 2.** Insomnia prevalence using different definitions.

| | Frequency (%) |
|---|---|
| Nighttime insomnia symptoms (*n* = 580) | |
| Yes | 189 (32.6) |
| No | 391 (67.8) |
| PSIQ Score (*n* = 552) | |
| >5 (poor sleep) | 359 (65.0) |
| ≤5 (good sleep) | 193 (35.0) |
| Insomnia Severity Index (*n* = 567) | |
| No insomnia (0–7) | 241 (42.5) |
| Sub-threshold insomnia (8–14) | 217 (38.3) |
| Moderate insomnia (15–21) | 99 (17.4) |
| Severe insomnia (22–28) | 10 (1.8) |

Table 3 depicts descriptive results and bivariable associations. More females (56% vs. 44%) and more younger people participated in this study. Most of them were overweight or obese. This study reported poor housing conditions with high prevalence of dampness (56.3%), visible mold (52.5%), moldy smell (50.2%), and crowding (72.8%). Depression (31.7%) and anxiety (32.6%) were common in this population. There was a high prevalence of "woke up during the night due to terrifying dreams, nightmares, or flashbacks related to a traumatic event" (47.3%).

**Table 3.** Associations between insomnia based on ISI score ≥15 and risk factors and odds ratio and 95% CI (*n* = 567).

| Variables | Total<br>*n* (%) | Insomnia | | Unadjusted<br>Odds Ratio<br>(95% CI) | *p* Value |
| --- | --- | --- | --- | --- | --- |
| | | Yes<br>(ISI Score ≥15)<br>*n* (%) | No<br>(ISI Score <15)<br>*n* (%) | | |
| Demographics | | | | | |
| Sex (*n* = 567) | | | | | |
| Male | 247 (43.6) | 40 (36.7) | 207 (45.2) | 0.70 (0.46, 1.08) | 0.109 |
| Female | 320 (56.4) | 69 (63.3) | 251 (54.8) | 1.00 | - |
| Age group, in years (*n* = 567) | | | | | |
| 18–29 | 175 (30.9) | 26 (23.9) | 149 (32.5) | 1.11 (0.49, 2.50) | 0.810 |
| 30–39 | 136 (24.0) | 30 (27.5) | 106 (23.1) | 1.79 (0.80, 4.04) | 0.159 |
| 40–49 | 92 (16.2) | 31 (28.4) | 61 (13.3) | 3.22 (1.41, 7.35) | 0.006 |
| 50–59 | 98 (17.3) | 13 (11.9) | 85 (18.6) | 0.97 (0.39, 2.41) | 0.945 |
| 60+ | 66 (11.6) | 9 (8.3) | 57 (12.4) | 1.00 | - |
| Body Mass Index (BMI) (*n* = 530) | | | | | |
| Obese | 245 (46.2) | 46 (46.0) | 199 (46.3) | 0.75 (0.45, 1.25) | 0.271 |
| Overweight | 149 (28.1) | 22 (22.0) | 127 (29.5) | 0.56 (0.31, 1.03) | 0.061 |
| Neither obese nor overweight | 136 (25.7) | 32 (32.0) | 104 (24.2) | 1.00 | - |
| Education level (*n* = 561) | | | | | |
| Less than secondary school graduation | 206 (36.7) | 37 (34.6) | 169 (37.2) | 0.80 (0.49, 1.33) | 0.391 |
| Secondary school graduation | 173 (30.8) | 31 (29.0) | 142 (31.3) | 0.80 (0.47, 1.35) | 0.407 |
| Some university/completed university/technical school | 182 (32.4) | 39 (36.4) | 143 (31.5) | 1.00 | - |
| Employment status (*n* = 553) | | | | | |
| Social assistance/unemployment insurance | 130 (23.5) | 31 (29.0) | 99 (22.2) | 1.74 (0.97, 3.13) | 0.064 |
| Unemployed | 144 (26.0) | 27 (25.2) | 117 (26.2) | 1.28 (0.71, 2.33) | 0.413 |
| Other including retired or home makers | 115 (20.8) | 24 (22.4) | 91 (20.4) | 1.47 (0.79, 2.72) | 0.226 |
| Employed (full-time, part-time, self-employed) | 164 (29.7) | 25 (23.4) | 139 (31.2) | 1.00 | - |
| Money left at the end of the month (*n* = 562) | | | | | |
| Not enough money | 328 (58.4) | 75 (68.8) | 253 (55.8) | 1.66 (0.93, 2.95) | 0.086 |
| Just enough money | 122 (21.7) | 17 (15.6) | 105 (23.2) | 0.91 (0.44, 1.87) | 0.787 |
| Some money | 112 (19.9) | 17 (15.6) | 95 (21.0) | 1.00 | - |
| Attend residential school (*n* = 567) | | | | | |
| Yes | 191 (33.7) | 45 (41.3) | 146 (31.9) | 1.50 (0.98, 2.31) | 0.063 |
| No | 376 (66.3) | 64 (58.7) | 312 (68.1) | 1.00 | |
| Parents or grandparents attend a residential school (*n* = 567) | | | | | |
| Yes | 490 (86.4) | 98 (89.9) | 392 (85.6) | 1.09 (0.49, 2.43) | 0.826 |
| No | 34 (6.0) | 3 (2.8) | 31 (6.8) | 0.42 (0.10, 1.74) | 0.233 |
| Do not know | 43 (7.6) | 8 (7.3) | 35 (7.6) | 1.00 | - |
| Life-style factors | | | | | |
| Smoking status (*n* = 563) | | | | | |
| Current smoker | 408 (72.5) | 93 (85.3) | 315 (69.4) | 2.42 (1.21, 4.86) | 0.013 |
| Ex-smoker | 63 (11.2) | 6 (5.5) | 57 (12.6) | 0.86 (0.29, 2.51) | 0.787 |
| Never smoker | 92 (16.3) | 10 (9.2) | 82 (18.1) | 1.00 | - |

**Table 3.** *Cont.*

| Variables | Total | Insomnia | | Unadjusted Odds Ratio (95% CI) | *p* Value |
|---|---|---|---|---|---|
| | *n* (%) | Yes (ISI Score ≥15) *n* (%) | No (ISI Score <15) *n* (%) | | |
| Marijuana use (*n* = 563) | | | | | |
| Regularly | 163 (29.0) | 28 (25.9) | 135 (29.7) | 0.92 (0.56, 1.52) | 0.757 |
| Occasionally | 89 (15.8) | 23 (21.3) | 66 (14.5) | 1.55 (0.89, 2.70) | 0.120 |
| No use | 311 (55.2) | 57 (52.8) | 254 (55.8) | 1.00 | - |
| Alcohol consumption per week (*n* = 398) | | | | | |
| More than 1 per week | 155 (38.9) | 32 (43.2) | 123 (38.0) | 1.33 (0.72, 2.46) | 0.370 |
| One-per week | 121 (30.4) | 22 (29.7) | 99 (30.5) | 1.13 (0.58, 2.21) | 0.710 |
| Non-drinker | 122 (30.7) | 20 (27.0) | 102 (31.5) | 1.00 | - |
| Non-medical drugs (*n* = 564) | | | | | |
| Yes | 35 (6.2) | 11 (10.2) | 24 (5.3) | 2.04 (0.97, 4.31) | 0.061 |
| No | 529 (93.8) | 97 (89.8) | 432 (94.7) | 1.00 | - |
| Physical activities at least 3 weeks (*n* = 528) | | | | | |
| Yes | 290 (54.9) | 48 (47.1) | 242 (56.8) | 0.68 (0.44, 1.04)) | 0.076 |
| No | 238 (45.1) | 54 (52.9) | 184 (43.2) | 1.00 | - |
| Screen time-2 h or less (*n* = 436) | | | | | |
| Yes | 289 (66.3) | 66 (71.0) | 223 (65.0) | 1.31 (0.80, 2.17) | 0.282 |
| No | 147 (33.7) | 27 (29.0) | 120 (35.0) | 1.00 | - |
| Average number of caffeinated drinks per day (*n* = 564) | | | | | |
| >5 per day | 131 (23.2) | 30 (27.8) | 101 (22.1) | 1.95 (0.86, 4.38) | 0.107 |
| 2–5 per day | 272 (48.2) | 45 (41.7) | 227 (49.8) | 1.30 (0.60, 2.81) | 0.505 |
| 1 per day | 93 (16.5) | 24 (22.2) | 69 (15.1) | 2.28 (0.98, 5.29) | 0.055 |
| None | 68 (12.1) | 9 (8.3) | 59 (12.9) | 1.00 | |
| Health outcomes | | | | | |
| Number of chronic health conditions (*n* = 567) | | | | | |
| Three or more conditions | 181 (32.0) | 47 (43.1) | 134 (29.3) | 2.44 (1.37, 4.33) | 0.002 |
| Two conditions | 91 (16.0) | 22 (20.2) | 69 (15.1) | 2.22 (1.13, 4.33) | 0.020 |
| One condition | 136 (24.0) | 20 (18.3) | 116 (25.3) | 1.19 (0.62, 2.33) | 0.595 |
| None | 159 (28.0) | 20 (18.3) | 139 (30.3) | 1.00 | - |
| Physical health (*n* = 566) | | | | | |
| Poor | 56 (9.9) | 23 (21.1) | 33 (7.2) | 5.58 (2.05, 15.18) | 0.001 |
| Fair | 121 (21.4) | 35 (32.1) | 86 (18.8) | 3.26 (1.28, 8.29) | 0.013 |
| Good | 243 (42.9) | 30 (27.5) | 213 (46.6) | 1.13 (0.44, 2.86) | 0.802 |
| Very Good | 92 (16.3) | 15 (13.8) | 77 (16.8) | 1.56 (0.57, 4.29) | 0.391 |
| Excellent | 54 (9.5) | 6 (5.5) | 48 (10.5) | 1.00 | - |
| Mental health (*n* = 563) | | | | | |
| Poor | 31 (5.5) | 10 (9.3) | 21 (4.6) | 3.65 (1.31, 10.17) | 0.013 |
| Fair | 99 (17.6) | 31 (28.7) | 68 (14.9) | 3.49 (1.55, 7.89) | 0.003 |
| Good | 217 (38.5) | 42 (38.9) | 175 (38.5) | 1.84 (0.85, 3.98) | 0.122 |
| Very Good | 138 (24.5) | 16 (14.8) | 122 (26.8) | 1.00 (0.42, 2.39) | 0.990 |
| Excellent | 78 (13.9) | 9 (8.3) | 69 (15.2) | 1.00 | - |

**Table 3.** *Cont.*

| Variables | Total | Insomnia | | Unadjusted Odds Ratio (95% CI) | *p* Value |
|---|---|---|---|---|---|
| | *n* (%) | Yes (ISI Score ≥15) *n* (%) | No (ISI Score <15) *n* (%) | | |
| Depression (*n* = 527) | | | | | |
| Yes | 167 (31.7) | 52 (51.5) | 115 (27.0) | 2.87 (1.84, 4.48) | <0.0001 |
| No | 360 (68.3) | 49 (48.5) | 311 (73.0) | 1.00 | - |
| Anxiety (*n* = 528) | | | | | |
| Yes | 172 (32.6) | 51 (51.0) | 121 (28.3) | 2.64 (1.69, 4.12) | <0.0001 |
| No | 356 (67.4) | 49 (49.0) | 307 (71.7) | 1.00 | - |
| Post-traumatic stress disorder (*n* = 531) | | | | | |
| Yes | 58 (10.9) | 21 (21.4) | 37 (8.5) | 2.92 (1.62, 5.26) | <0.0001 |
| No | 473 (89.1) | 77 (78.6) | 396 (91.5) | 1.00 | - |
| Chronic pain (*n* = 546) | | | | | |
| Yes | 128 (23.4) | 46 (43.8) | 82 (18.6) | 3.41 (2.17, 5.38) | <0.0001 |
| No | 418 (76.6) | 59 (56.2) | 359 (81.4) | 1.00 | - |
| Prescription medication use for any health condition (*n* = 561) | | | | | |
| Yes | 251 (44.7) | 63 (57.8) | 188 (41.6) | 1.92 (1.26, 2.94) | 0.002 |
| No | 310 (55.3) | 46 (42.2) | 264 (58.4) | 1.00 | - |
| Housing conditions | | | | | |
| Dampness (*n* = 563) | | | | | |
| Yes | 317 (56.3) | 68 (62.4) | 249 (54.8) | 1.36 (0.89, 2.10) | 0.155 |
| No | 246 (43.7) | 41 (37.6) | 205 (45.2) | 1.00 | - |
| Visible mold (*n* = 562) | | | | | |
| Yes | 295 (52.5) | 68 (62.4) | 227 (50.1) | 1.65 (1.08, 2.54) | 0.022 |
| No | 267 (47.5) | 41 (37.6) | 226 (49.9) | 1.00 | - |
| Moldy smell (*n* = 562) | | | | | |
| Yes | 283 (50.2) | 65 (59.6) | 218 (47.9) | 1.65 (1.07, 2.54) | 0.022 |
| No | 281 (49.8) | 44 (40.4) | 237 (52.1) | 1.00 | - |
| Smoke inside house (*n* = 561) | | | | | |
| Yes | 243 (43.3) | 49 (45.8) | 194 (42.7) | 1.13 (0.74, 1.73) | 0.565 |
| No | 318 (56.7) | 58 (54.2) | 260 (57.3) | 1.00 | - |
| Crowding (*n* = 556) | | | | | |
| >1 person/bedroom | 405 (72.8) | 81 (75.7) | 324 (72.2) | 1.20 (0.74, 1.96) | 0.460 |
| <= person/bedroom | 151 (27.2) | 26 (24.3) | 125 (27.8) | 1.00 | - |
| Sleeping conditions | | | | | |
| Place of sleep (*n* = 442) | | | | | |
| Bedroom | 330 (74.7) | 66 (69.5) | 264 (76.1) | 0.83 (0.22, 3.11) | 0.786 |
| Living room | 52 (11.8) | 12 (12.6) | 40 (11.5) | 1.00 (0.24, 4.23) | 1.000 |
| Basement | 47 (10.6) | 14 (14.7) | 33 (9.5) | 1.41 (0.34, 5.93) | 0.636 |
| Other | 13 (2.9) | 3 (3.2) | 10 (2.9) | 1.00 | - |

**Table 3.** *Cont.*

| Variables | Total | Insomnia | | Unadjusted Odds Ratio (95% CI) | *p* Value |
|---|---|---|---|---|---|
| | *n* (%) | Yes (ISI Score ≥15) *n* (%) | No (ISI Score <15) *n* (%) | | |
| Sleeping arrangement shared with (*n* = 560) | | | | | |
| Child | 116 (20.7) | 29 (26.6) | 87 (19.3) | 2.26 (1.00, 5.11) | 0.050 |
| Spouse or partner | 177 (31.6) | 38 (34.9) | 139 (30.8) | 1.85 (0.84, 4.07) | 0.124 |
| Alone | 197 (35.2) | 33 (30.3) | 164 (36.4) | 1.36 (0.62, 3.02) | 0.443 |
| Family member/Other | 70 (12.5) | 9 (8.3) | 61 (13.5) | 1.00 | - |
| Afraid to sleep at home (*n* = 566) | | | | | |
| Yes | 58 (10.2) | 26 (23.9) | 32 (7.0) | 4.16 (2.36, 7.34) | <0.0001 |
| No | 508 (89.8) | 83 (76.1) | 425 (93.0) | 1.00 | - |
| Feel safe to sleep at home (*n* = 565) | | | | | |
| Yes | 515 (91.2) | 91 (83.5) | 424 (93.0) | 0.38 (0.21, 0.71) | 0.002 |
| No | 50 (8.8) | 18 (16.5) | 32 (7.00 | 1.00 | - |
| Wake up during the night due to dreams, nightmares, or flashbacks related to a traumatic event (*n* = 564) | | | | | |
| Yes | 267 (47.3) | 73 (67.6) | 194 (42.5) | 2.82 (1.81, 4.39) | <0.0001 |
| No | 297 (52.7) | 35 (32.4) | 262 (57.5) | 1.00 | - |
| Taking medicine used for sleep (*n* = 560) | | | | | |
| Yes | 66 (11.8) | 25 (23.1) | 41 (9.1) | 1.92 (1.26, 2.94) | 0.002 |
| No | 494 (88.2) | 83 (76.9) | 411 (90.9) | 1.00 | - |
| Taking traditional medicine used for sleep (*n* = 561) | | | | | |
| Yes | 32 (5.7) | 14 (12.8) | 18 (4.0) | 3.55 (1.71, 7.39) | 0.001 |
| No | 529 (94.3) | 95 (87.2) | 434 (96.0) | 1.00 | - |
| Other aids used for sleep (*n* = 442) | | | | | |
| Yes | 55 (12.4) | 16 (16.8) | 39 (11.2) | 1.60 (0.85, 3.01) | 0.145 |
| No | 387 (87.6) | 79 (83.2) | 308 (88.8) | 1.00 | - |

An increased prevalence of insomnia was observed in those that were in the age group of 40–49 years and currently engaged in non-traditional use of tobacco (current smoker), who reported poor to fair physical and mental health, and with the presence of any of the following conditions: depression, anxiety, chronic pain, post-traumatic stress disorder, two or more chronic health conditions, or were living housing conditions with visible mold and moldy smell.

Prevalence of prescription medication use for any health condition among participants was 44.7%. In addition, more participants with insomnia had used a prescribed sleep medication (57.8% vs. 41.6%), over-the counter sleep aids (23.1% vs. 9.1%), and traditional medicines (12.8% vs. 4.0%) compared with those who did not have insomnia in this population.

The prescription medication use for any health condition and other medication used for sleep were associated with increased insomnia. Those who were afraid to sleep in their own house and woke up during the night due to terrifying dreams, nightmares, or flashbacks related to a traumatic event had higher odds of insomnia. If participants felt safe to sleep in their house, this had a protective effect on the prevalence of insomnia.

Multivariate logistic regression results are presented in Table 4. Observed bivariable associations with age (40–49 years age group), physical health (poor to fair), presence

with depression, chronic pain, prescription medication use for any health condition, and waking up during the night due to terrifying dreams, nightmares, or flashbacks related to a traumatic event and insomnia were retained in the final multivariable model.

**Table 4.** Multivariable Associations between insomnia based on ISI score $\geq$15 and risk factors and odds ratio and 95% CI (*n* = 567).

| Variables | Adjusted Odds Ratio (95% CI) | *p* Value |
|---|---|---|
| Demographics | | |
| Sex | | |
| Male | 0.84 (0.50, 1.41) | 0.505 |
| Female | 1.00 | - |
| Age group, in years | | |
| 18–29 | 2.58 (0.84, 7.89) | 0.096 |
| 30–39 | 2.42 (0.85, 6.95) | 0.099 |
| 40–49 | 5.65 (1.97, 16.15) | 0.001 |
| 50–59 | 0.93 (0.32, 2.68) | 0.888 |
| 60+ | 1.00 | - |
| Health Outcomes | | |
| Number of chronic health conditions | | |
| Three or more conditions | 1.02 (0.45, 2.31) | 0.956 |
| Two conditions | 1.79 (0.81, 3.99) | 0.152 |
| One condition | 0.88 (0.40, 1.95) | 0.752 |
| None | 1.00 | - |
| Physical health | | |
| Poor | 6.67 (1.87, 23.82) | 0.003 |
| Fair | 3.47 (1.07, 11.22) | 0.038 |
| Good | 1.18 (0.37, 3.79) | 0.776 |
| Very Good | 2.20 (0.63, 7.60) | 0.214 |
| Excellent | 1.00 | - |
| Depression | | |
| Yes | 1.72 (0.99, 3.01) | 0.056 |
| No | 1.00 | - |
| Chronic pain | | |
| Yes | 1.91 (1.00, 3.64) | 0.050 |
| No | 1.00 | - |
| Prescription medication use for any health condition | | |
| Yes | 1.90 (1.02, 3.56) | 0.043 |
| No | 1.00 | - |
| Sleeping conditions | | |
| Wake up during the night due to dreams, nightmares or flashbacks related to a traumatic event | | |
| Yes | 2.20 (1.29, 3.76) | 0.004 |
| No | 1.00 | - |

## 4. Discussion

The prevalence of insomnia (ISI score ≥15) among participants in the two Cree First Nation communities in Saskatchewan was 19.2%. According to the definition of nighttime insomnia symptoms, the prevalence of insomnia was 32.6%, a finding similar to the prevalence (25% to 33% and 33.7%) reported for North American Indian/Alaska Native populations [35,36]. According to the ISI definition, the prevalence of insomnia was lower (19.2%) compared to North American Indian/Alaska Native populations, but within the range (13.4% to 38.0%) of the general population in Canada [5,8,11,21,22,35].

The results of this study demonstrate that age, physical health, being diagnosed with depression and chronic pain; as well as prescription medication use for any health condition were related to increased prevalence of insomnia in two Saskatchewan First Nation communities; findings that had previously been shown in other studies and populations [2,21,26,54]. There was a high prevalence of waking up during the night due to terrifying dreams, nightmares, or flashbacks related to a traumatic event among participants, an important finding from the populations in two First Nation communities. Additionally, it was observed that insomnia was associated with 40–49 years middle-age group rather than in the older age group in this study.

Studies have shown that the prevalence of insomnia increases with age [2,6,21,25,26,54]. In Western countries, insomnia is an important health issue, as most of the older people have reported having insomnia [55–58]. In contrast, this population had a higher prevalence of insomnia in the age group that was 40–49 years. One of the reasons could be that there is a smaller proportion of older people residing in the two First Nation communities, as this study had a small number of participants that were in the older age group (*n* = 66, who are 60 years and older). According to the 2016 Canadian Census, the proportion of 65 years of age and older adults was 6.4% of the total First Nations population [59].

Previous studies have shown many chronic health conditions are related to insomnia [10,26–28,60–62]. Authors have reported that heart disease [10,26,60,61], angina [28], hypertension [10,28,60], asthma [10,28], diabetes [10,28,61], stroke [28], obstructive pulmonary disease [10,26,28,60,61], back problems [26], hip impairment [26], arthritis [10,28], urinary problems [60,61], prostate problems [26], bowel disorder [27], gastrointestinal problems [10,60], sleep apnea [29,30], kidney disease [31,32], and thyroid disorders [33] have been associated with an increased risk of insomnia. Koyanagi et al. [28] reported that severe or extreme insomnia symptoms such as falling asleep, waking up frequently during the night, or waking up too early in the morning were associated with chronic health conditions. In addition, Koyanagi et al. [28] and Budhiraja et al. [10] observed that there was a significant linear dose-dependent relationship between the number of chronic conditions and sleep problems including insomnia. Similarly, this study also observed a significant dose-dependent relationship between the number of chronic conditions and insomnia in the univariate analysis (three or more conditions: OR = 2.44, 95% CI: 1.37, 4.33; two conditions: OR = 2.22, 95% CI: 1.13, 4.33), but the significance disappeared in the multivariate analysis.

Many common medications for chronic health conditions may have side effects such as sleepiness, daytime drowsiness, and being awake all night with insomnia [63]. Studies have shown that medications taken for any chronic health conditions such as antihypertensives, sympathomimetics, xanthine derivatives, and antidepressants are causes of insomnia [14,26,64–66]. Prescription medications taken to treat chronic conditions, non-prescription drugs (over-the-counter), and substance misuse can interfere with sleep resulting in severe insomnia as a side effect [28,67,68]. Similar findings were observed in this study. Participants taking prescription medications had an increased risk of insomnia, supported by the above claims. In addition, more participants with insomnia had used a prescribed sleep medication, over-the counter sleep aids, and traditional medicines compared with those who did not have insomnia in this popultaion. These results suggest that many people may have tried to overcome the problem by self-medicating.

In these two Saskatchewan Cree First Nation communities, chronic pain was a common condition (23.4%) and a significant risk factor for insomnia. Similarly, studies have shown persons with chronic pain have statistically higher levels of insomnia than those without chronic pain [54,60]. Of those with chronic pain, an estimated 50–80% have ongoing sleep difficulties [69]. About 50% of individuals with back pain indicated that disrupted sleep leads to the exacerbation of chronic back pain [70,71]. According to a recent Health Canada Report [72], it was estimated that approximately one in five Canadians live with chronic pain, and chronic pain was reported to be more severe in women and in Indigenous peoples in Canada [72].

Consistent with previous reports [2,21,34,55,73,74], fair to poor self-rated physical health was associated with a higher risk of insomnia in these two First Nation communities. Fair to poor self-rated physical health was common in these communities and about one-third (31.3%) of participants reported fair to poor physical health. Depression was prevalent (31.7%) in these communities as well. It was also found that participants with depression were 1.72 times more likely to have insomnia compared to those who did not ($p$ = 0.056). This was consistent with the established relationship between depression and insomnia [2,25,26,28,35]. IsHak et al. [75] reported that people with chronic pain tend to have higher rates of depression, but the direction of the relationship between the two is not clear.

Although not significant in the multivariate model, the bivariable model showed a strong association between insomnia and "being afraid to sleep at home" and "not feeling safe at home". This suggests that "fear to sleep at home" was an important contributor to insomnia in this First Nations population. In turn, this study identified a strong relationship (OR = 2.20; 95% CI: 1.29, 3.76) between waking up during the night due to terrifying dreams, nightmares, or flashbacks related to a traumatic event and insomnia. This was an important finding in this First Nations population and it has not been previously reported in literature. These terrified dreams, nightmares, or flashbacks can cause significant distress both during and after awakening, and may occur several times a week [76] resulting in insomnia [77,78]. Other authors reported an association between insomnia and PTSD [35,77,79]. Although a bivariable relationship with PTSD and insomnia (OR = 2.92; 95% CI: 1.62, 5.26) was observed in this study, the results did not remain significant after adjusting for other factors in the multivariate analysis.

Sex differences found in insomnia were related to biological, psychological, and social factors [23]. Insomnia prevalence has been reported to be about 1.5 times more common in women than in men [24]. Another study reported that the prevalence of insomnia symptoms and the subtypes was more prevalent in women than men [16]. In contrast to these studies, this study did not identify such a difference in relation to sex.

Poor housing conditions continue to be a major problem in First Nation communities in Canada [80,81]. In an earlier Canadian study with two First Nation communities, First Nations' houses were shown to have poor ventilation and presence of mold [82]. This study reported poor housing conditions with high prevalence of dampness, visible mold, moldy smell, and crowding. Janson et al. [83] reported that living in houses with signs of building dampness, including water damage, visible mold, and floor dampness was an independent risk factor for insomnia and insomnia related symptoms. Tiesler et al. [84] reported that exposure to visible mold or dampness in the house increased the risk for sleep problems including insomnia symptoms in children. The bivariable analysis in this study suggests that visible mold and moldy smell were independent risk factors for insomnia but did not reach significance in the multivariate model.

The health disparities of Indigenous people are related to structural (historical, political, social, and economic) determinants of health [85]. First Nations perspectives on health and wellness reflects the connection between physical, mental, emotional, and spiritual dimensions of wellbeing [86]. Colonialism has systematically severed links between these dimensions in assimilation efforts that continue to underpin health inequities between Indigenous and non-Indigenous peoples in Canada [87]. The residential school experience,

for example, negatively impacts the health and wellbeing of survivors, as well as their children and grandchildren. Until the mid-1990s, children as young as age four were forcibly removed from their families to live in and attend these schools that stripped them of their language, culture, customs, and relationships, intentionally severing the ties to their families and communities to assimilate them into settler society. Children in these schools experienced mental, physical, and sexual abuse and suffered from malnutrition and infectious diseases such as tuberculosis [56,85,88]. Notably, nighttime in residential schools was a time during which children were especially vulnerable to these abuses. The significant finding of the association between terrified dreams, nightmares, and flashbacks, accompanied by the bivariable significant association between residential school attendance and insomnia, is compelling evidence of the ongoing impacts of colonization. This merits further investigation.

This study has a number of strengths including the number of participants engaged in the project; detailed information about sleep; sleep duration and insomnia; environmental housing conditions; medical history; the strength of associations between age, physical health, depression, chronic pain, and bad dreams; and prescription medication use for any health condition with insomnia in these unique communities. The common experience of colonization and the associations we observed in this study are compelling. Several limitations exist in this study. First, as these were cross-sectional observations, causation can only be speculative. Another potential limitation is that this study is based on self-reported data, and therefore some recall bias may exist. Another limitation is that there are no standard definitions for insomnia in epidemiological research for insomnia [8,28]. Although a seven-item set of questions to define insomnia and trouble going to sleep or staying asleep as nighttime insomnia symptoms was used, other objective clinical diagnostic criteria exist for the diagnosis of insomnia including the Diagnostic and Statistical Manual of Mental Disorders, Fourth Edition (DSM-IV) [89], the International Classification of Sleep Disorders (ICSD) [90], and the International Classification of Diseases (ICD-10) [91]. Many studies have used the positive responses to general questions about difficulty initiating or maintaining sleep [92]. To facilitate comparability this study reported findings consistent with a range of definitions and measures used in the literature.

## 5. Conclusions

In two Cree First Nation communities in Saskatchewan, Canada, we observed about one fifth of the participants reported having insomnia. Associated risk factors were age, poor to fair physical health, being diagnosed with depression, chronic pain, prescription medication use for any health condition, and sleep disruption related to terrifying dreams, nightmares, or flashbacks related to a traumatic event. The significant association between terrifying dreams, nightmares, or flashbacks related to a traumatic event and insomnia was an important finding in this study, which was impacted by colonization. To explore this further, a qualitative research study will be conducted. It is important to note that the prevalence of insomnia was comparable with rest of the Canadian population (mainly the population of European descent); however, insomnia was more common in the middle age groups in this population. Identifying these associations provides directions for future research in this area. Additionally, this unique study, which engaged two similar but unique First Nation communities, will assist health care providers in diagnosing and treating patients with insomnia in these communities.

**Author Contributions:** Conceptualization: J.AD., S.A., M.K., P.P., D.CR., S.K., N.K., M.F., R.S., and the First Nations Sleep Health Project Team; data curation: B.PR. and K.M.; formal analysis: C.PK.; funding acquisition: J.AD., S.A., M.K., and P.P.; investigation: J.AD., S.A., M.F., M.K., and P.P.; methodology: J.A.D., P.P., S.A., M.K., C.P.K., and M.F.; project administration: P.P.; resources: J.S., C.B., R.S., M.F., and T.S.-W.; supervision: J.A.D. and P.P.; visualization: S.A., J.S., C.B., and V.RR.; writing—original draft: J.A.D., C.PK., and P.P.; writing—review and editing: J.AD., C.PK., K.M., S.A., D.CR., M.K., S.K., N.K., J.S., C.B., V.RR., M.F., R.S., B.PR., T.S.-W., and P.P. All authors have read and agreed to the published version of the manuscript.

**Funding:** This research was funded by a grant from the Canadian Institutes of Health Research—"Assess, Redress, Re-assess: Addressing Disparities in Sleep Health among First Nations People", CIHR MOP-391913-IRH-CCAA-11829-FRN PJT-156149.

**Institutional Review Board Statement:** The study was conducted according to the guidelines of the Declaration of Helsinki, and approved by the Biomedical Research Ethics Board of University of Saskatchewan (Bio #18-110 and date of approval: 21 June 2018).

**Informed Consent Statement:** Written informed consent has been obtained from all participants involved in the study.

**Acknowledgments:** The First Nations Sleep Health Project Team consists of: James A Dosman, (Designated Principal Investigator, University of Saskatchewan, Saskatoon, SK Canada); Punam Pahwa, (Co-Principal Investigator, University of Saskatchewan, Saskatoon SK Canada); Malcolm King, (Co-Principal Investigator, University of Saskatchewan, Saskatoon, SK Canada), Sylvia Abonyi, (Co-Principal Investigator, University of Saskatchewan, Saskatoon, SK Canada); Co-Investigators: Mark Fenton, Chandima P Karunanayake, Shelley Kirychuk, Niels Koehncke, Joshua Lawson, Robert Skomro, Donna Rennie, Darryl Adamko; Collaborators: Roland Dyck, Thomas Smith-Windsor, Kathleen McMullin, Rachana Bodani, John Gjerve, Bonnie Janzen and Vivian R Ramsden, Gregory Marchildon, Kevin Colleaux. Project Manager: Brooke P Russell; Community Partners: Jeremy Seeseequasis, Clifford Bird; Roy Petit; Edward Henderson; Raina Henderson; Dinesh Khadka. We are grateful for the contributions from Elders and community leaders that facilitated the engagement necessary for the study, and all participants.

**Conflicts of Interest:** The authors declare no conflict of interest. The funders had no role in the design of the study; in the collection, analyses, or interpretation of data; in the writing of the manuscript; or in the decision to publish the results.

## Abbreviations

| | |
|---|---|
| CAD | Canadian |
| ISI | Insomnia Severity Index |
| PSQI | Pittsburgh Sleep Quality Index |
| FNSHP | First Nations Sleep Health Project |
| SD | Standard Deviation |
| OR | Odds Ratio |
| CI | Confidence Interval |
| DSM-IV | Diagnostic and Statistical Manual of Mental Disorders |
| ICSD | International Classification of Sleep Disorders |
| ICD | International Classification of diseases |

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
