# Peer review of "Prevalence of Insomnia in Two Saskatchewan First Nation Communities"

_2624-5175, doi:10.3390/clockssleep3010007_

Round 1
Reviewer 1 Report
The authors examined the prevalence and risk factors for insomnia in two Cree First Nation communities in Saskatchewan, Canada. The present study found that the prevanlece of insomnia in indigenous Canadian was estimated as 19.2% using Insomnia Severity Index score, and as 32.6% as defined by nighttie insomnia symptoms, which seems to be similar to the insomnia prevalence in Canada (ranging 13.4% to 38%). A lot of risk factors for insomnia including middle age, poor to fair physical health, concomitant mental disorder and chronic pain were detected by multivariate logistic regression analysis. Of importance, their traumatic events (colonization and/or bullying/abuse in family and residential school) may be associated with imsomniac symptoms, such as waking up during the night due to terrifying dreams, nightmares, or flashbacks, which were another risk factors for insomnia.
The findings obtained by this study are of interest. However, several points need clarifying. These are given below.
Major comments:
1. In the “Introduction” section, the authors should explain the reason why the present study focused on two Cree First Nation communities in Saskatchewan, Canada. I think that all the researchers are not familiar to the history and status quo of indigenous Canadian. For example, a recent paper described by TransFORmation of IndiGEnous PrimAry HEAlthcare delivery (FORGE AHEAD) program (2020, PMID: 32045618) has noted, “the health status of Indigenous peoples in Canada remains below that of the general population due to a range of historical, political, cultural, geographic and jurisdictional factors as well as intergenerational poverty, low levels of education, high unemployment, poor housing, high rates of depression and stress, negative stereotyping and stigmatization”. As described by the authors, in lines 199 to 213, the indigenous Canadian are frequently bullied and abused in youth and adult, which is important background investigating their prevalence of insomnia.
2. In the “Materials and Methods” section, the authors noted, “Written consent was obtained from individual 244 participants in this research collaboration with the two Saskatchewan First Nation communities (lines 244 to 245)”. But it is appopriate to specify whether “informed consent” was obtained from each participant after sufficient explanation and information of the current study.
In connection with this, I wonder that the cognitive function is normal in all the participants. Because elder people are susceptible to cognitive impairment, it is of importance that 60 years and older adults have suitable cognition to understand the explanation of research.
3. In lines 144 to 146, the authors mentioned that several prescribed medications (e.g,, antihypertensives, sympthomimetics, xanthine derivatives, and antidepressants) were causes of insomnia. However, I don’t understand what the parameter, “prescription medication use” is pointing out. In lines 260 to 263, several health condions were listed and the number of chronic health conditions was collected in the present study, Considering the collection of data, I guess that “prescription medication use” means “prescription medication use for the above-described chronic health conditions”. Or “all prescribed medication use for any health conditions” were collected as study parameters? Please clarify it.
Minor comments:
Many typographical errors are observed. For example, space is excess or sparse (many positions), “PSIQ” (line 82 and Table 2), “p=<0.0001” (lines 86 and 88, I think “=” should be excluded). Please correct carefully and strictly.
I hope these comments will be helpful.
Author Response
Reviewer 1
Comments and Suggestions for Authors
The authors examined the prevalence and risk factors for insomnia in two Cree First Nation communities in Saskatchewan, Canada. The present study found that the prevalence of insomnia in indigenous Canadian was estimated as 19.2% using Insomnia Severity Index score, and as 32.6% as defined by nighttime insomnia symptoms, which seems to be similar to the insomnia prevalence in Canada (ranging 13.4% to 38%). A lot of risk factors for insomnia including middle age, poor to fair physical health, concomitant mental disorder and chronic pain were detected by multivariate logistic regression analysis. Of importance, their traumatic events (colonization and/or bullying/abuse in family and residential school) may be associated with insomniac symptoms, such as waking up during the night due to terrifying dreams, nightmares, or flashbacks, which were another risk factors for insomnia.
The findings obtained by this study are of interest. However, several points need clarifying. These are given below.
Major comments:
Comment #1: In the “Introduction” section, the authors should explain the reason why the present study focused on two Cree First Nation communities in Saskatchewan, Canada. I think that all the researchers are not familiar to the history and status quo of indigenous Canadian. For example, a recent paper described by TransFORmation of IndiGEnous PrimAry HEAlthcare delivery (FORGE AHEAD) program (2020, PMID: 32045618) has noted, “the health status of Indigenous peoples in Canada remains below that of the general population due to a range of historical, political, cultural, geographic and jurisdictional factors as well as intergenerational poverty, low levels of education, high unemployment, poor housing, high rates of depression and stress, negative stereotyping and stigmatization”. As described by the authors, in lines 199 to 213, the indigenous Canadian are frequently bullied and abused in youth and adult, which is important background investigating their prevalence of insomnia.
Response to Comment #1: Importance of the present study on First Nations insomnia prevalence was added to introduction. Please see below. Lines 68-74.
There are health inequities between Indigenous and non-Indigenous (mainly descending from Europeans) peoples in Canada [37-39] located in historical, political, social, and economic conditions that inequitably influence Indigenous health. Colonization is a key upstream determinant of health for Indigenous peoples, that impacts more proximal determinants of Indigenous health through systemic and individual racism [40-43]. One consequence is that Indigenous youth and adults experience bullying and abuse [44-45], that affect sleep and are linked to the prevalence of insomnia.
Comment #2: In the “Materials and Methods” section, the authors noted, “Written consent was obtained from individual participants in this research collaboration with the two Saskatchewan First Nation communities (lines 278 to 279)”. But it is appropriate to specify whether “informed consent” was obtained from each participant after sufficient explanation and information of the current study.
Response to Comment #2: This statement is corrected. “In addition to community consent achieved through the collaboration process, informed written consent was obtained from each individual participant following a discussion of study objectives, procedures, risks, and benefits.” Please see lines: 90-92.
Comment #3: In connection with this, I wonder that the cognitive function is normal in all the participants. Because elder people are susceptible to cognitive impairment, it is of importance that 60 years and older adults have suitable cognition to understand the explanation of research.
Response to Comment #3: These are community dwelling older adults, a number are recognized knowledge keepers. Just like everyone else, they came to participate in the study -getting themselves to the health and/or youth centres where data collection took place. We had absolutely no indications from our community partners, not from our community Research Assistants that cognitive deficits in older people during the consent process.
Comment #4: In lines 144 to 146, the authors mentioned that several prescribed medications (e.g,, antihypertensives, sympthomimetics, xanthine derivatives, and antidepressants) were causes of insomnia. However, I don’t understand what the parameter, “prescription medication use” is pointing out. In lines 260 to 263, several health conditions were listed and the number of chronic health conditions was collected in the present study, Considering the collection of data, I guess that “prescription medication use” means “prescription medication use for the above-described chronic health conditions”. Or “all prescribed medication use for any health conditions” were collected as study parameters? Please clarify it.
Response to Comment #4: Thank you for the comment. This variable is corrected to “prescription medication use for any health condition”. See highlighted text throughout the text.
Minor comments:
Comment #5: Many typographical errors are observed. For example, space is excess or sparse (many positions), “PSIQ” (line 82 and Table 2), “p=<0.0001” (lines 86 and 88, I think “=” should be excluded). Please correct carefully and strictly.
Response to Comment #5: Extra spaces deleted throughout the text and “=” signs were deleted from lines 187-189.
Reviewer 2 Report
Dear Authors
Could you specify which criteria was used for Table 3 Insomnia (yes or No)
Some of the discussion would have been better placed in the results sections.
In the Conclusion, the prevalance of insomnia was stated to be similar to other Canadian populations. This could have been expanded a little more and with additional information to highlight the differences in study population to “other” Canadian populations. This was stated earlier when referencing a study into Native American military population compared with non native American military population.
I would have preferred the Methods and Data collection to be mentioned before results were presented.
Author Response
Reviewer 2
Comments and Suggestions for Authors
Comment #1: Could you specify which criteria was used for Table 3 Insomnia (yes or No)
Response to Comment #1: Table 3 Insomnia (Yes or No) based on Insomnia Severity Index (ISI) score ≥15. This was mentioned in the methods section in lines 148-150. This information is added to the Table 3 & Table 4 headings and variable labels.
Comment #2: Some of the discussion would have been better placed in the results sections.
Response to Comment #2: Thank you for the comment. As reviewer suggested some of the discussion was placed back into the results sections. Please see lines: 200-205 and lines 211-214.
Comment #3: In the Conclusion, the prevalence of insomnia was stated to be similar to other Canadian populations. This could have been expanded a little more and with additional information to highlight the differences in study population to “other” Canadian populations. This was stated earlier when referencing a study into Native American military population compared with non-native American military population.
Response to Comment #3: “Other” population mainly consist of the population of European descent. There are differences in social and cultural structures within these two groups. Please see lines 371-372.
Comment #4: I would have preferred the Methods and Data collection to be mentioned before results were presented.
Response to Comment #4: According to “Clocks & Sleep” Journal publication guidelines, the Methods and Data collection sections should appear after the discussion section. After getting permission from the Assistant Editor, the Methods and Data Collection sections were moved to before the Results section.
Reviewer 3 Report
Overall this is interesting study for sleep topics
Sleep disorders are common in patients with thyroid disorders, kidney diseases, and obstructive sleep apnea.
The investigators should provide data on thyroid disorders, kidney diseases, and obstructive sleep apnea on tables.
Furthermore, citations should be provided in the introductions on those diseases
caffeine has been mentioned in the introduction, but data has not been provided, this is an important factor.
Author Response
Reviewer 3
Comments and Suggestions for Authors
Comment #1: Overall this is interesting study for sleep topics. Sleep disorders are common in patients with thyroid disorders, kidney diseases, and obstructive sleep apnea. The investigators should provide data on thyroid disorders, kidney diseases, and obstructive sleep apnea on tables. Furthermore, citations should be provided in the introductions on those diseases.
Response to Comment #1: The variable for number of chronic conditions were created using all positive responses to the following conditions: high blood pressure; heart problems; stroke; high cholesterol and/or triglycerides; diabetes; atrial fibrillation; chronic obstructive pulmonary disease/emphysema; asthma; chronic bronchitis; acid reflux; hypothyroidism; severe eyesight problem; sinus problems; Parkinson’s disease; sleep apnea; kidney disease; and restless legs syndrome.
We have investigated associations between variables of hypothyroidism, kidney disease, sleep apnea and insomnia as per request from reviewer #3 separately as shown in the Table below. The variables for hypothyroidism and kidney disease were not significant at univariate level thus did not qualify for the multivariate model. The sleep apnea variable was included into the multivariable model with other variables and it was observed that it was not significant at the 5% significance level. Therefore, there is no change in the multivariate model. These results were not included into tables separately as they were part of the “number of chronic conditions” variable. However, few citations were added to the introduction about the association between insomnia and thyroid disorders, kidney diseases, and obstructive sleep apnea.
|
Variables |
Insomnia
|
Unadjusted odds ratio (95% CI) |
P value |
adjusted odds ratio (95% CI) |
P value |
|
|
Yes (ISI Score ≥15) n(%) |
No (ISI Score <15) n (%) |
|||||
|
Chronic Condition |
|
|
|
|
|
|
|
Hypothyroidism |
|
|
|
|
|
|
|
Yes |
40 (36.7) |
207 (45.2) |
0.70 (0.46, 1.08) |
0.109 |
|
|
|
No |
69 (63.3) |
251 (54.8) |
1.00 |
- |
|
|
|
Kidney Disease |
|
|
|
|
|
|
|
Yes |
7 (6.8) |
13 (2.9) |
2.41 (0.94, 6.21) |
0.068 |
|
|
|
No |
96 (93.2) |
430 (97.1) |
1.00 |
- |
|
|
|
Sleep Apnea |
|
|
|
|
|
|
|
Yes |
13 (14.1) |
22 (5.2) |
2.99 (1.45, 6.20) |
0.003 |
2.20 (0.87, 5.54) |
0.094 |
|
No |
79 (85.9) |
401 (94.8) |
1.00 |
- |
|
|
Comment #2: Caffeine has been mentioned in the introduction, but data has not been provided, this is an important factor.
Response to Comment #2: Average number of caffeinated drinks per day data is added to Table 3. Please see page 8.
Round 2
Reviewer 1 Report
The authors have appropriately responded to my comment.
Their corrections have improved the scientific validity and soundness of this article.
Reviewer 2 Report
No changes required
Reviewer 3 Report
The authors have responded appropriately. This is an interesting paper with increased scientific soundness after their corrections. There are limitations to the study but they are stated.